# System Identification of an Aerial Delivery System with a Ram-Air Parachute Using a NARX Network

**Kemal Güven \*** and **Andaç Töre Şamiloğlu**

Mechanical Engineering Department, Başkent University, Ankara 06500, Turkey
\* Correspondence: kemalguven@baskent.edu.tr

**Abstract:** Neural networks are one of the methods used in system identification problems. In this study, a NARX network with a serial-parallel structure was used to identify an unknown aerial delivery system with a ram-air parachute. The dataset was created using the software-in-the-loop method (Software in the loop). Gazebo was used as the simulator and PX4 was used as the autopilot software. The performance of the NARX network differed according to parameters used, such as the selected training algorithm, input and output delays, the hidden layer, and the number of neurons. Within the scope of this study, each parameter was examined independently. Models were trained using MATLAB 2020a. The results demonstrated that the model with one hidden layer and five neurons, which was trained using the Bayesian regularization algorithm, was sufficient for this problem.

**Keywords:** Bayesian regularization; Levenberg–Marquardt; NARX network; scaled conjugate gradient; software in the loop

## 1. Introduction

In aircraft, system identification can be thought of as estimating aerodynamic parameters or defining a mathematical model of the system. Three methods have been proposed in the literature for the estimation of aerodynamic parameters of parachute landing systems [1]. The first of these covers analytical methods based on computational fluid dynamics. Others are wind tunnel tests and flight tests. In this study, we focused on the methods used in flight tests.

The purpose of system definition is to obtain a mathematical model according to the inputs and outputs obtained from the flight tests. Hamel and Jategaonkar proposed the 4M (maneuver, measurement, method, model—see Figure 1) requirements for successful system identification [2], arguing that:

- Control inputs should be created to cover extreme points;
- High-resolution measurements should be used;
- The possible mathematical model of the vehicle should be defined; and
- The most suitable method for the data should be chosen.

Jann and Strickert suggested separating the symmetric and asymmetric maneuvers that need to be carried out in the formation of data to be used in the definition process [3] (Figure 2).

The methods used in parameter estimation can be listed as the Equation-error, output-error, and filter-error methods. The question of which method to choose can be decided according to the measurement and the noise present in the process [2] (Figure 3). If disruptive factors can be ignored in both, the fastest method, the equality-error method, is preferred. If the disturbing factors are only assumed in the measurements, the output-error method is recommended, and if both are present, the filter-error method is recommended.

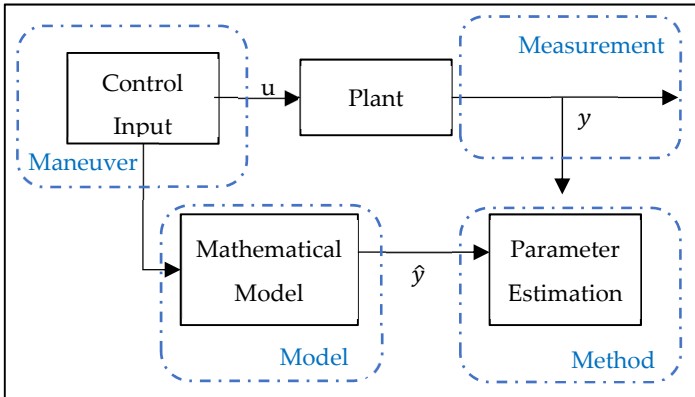

**Figure 1.** 4M-based system identification process.

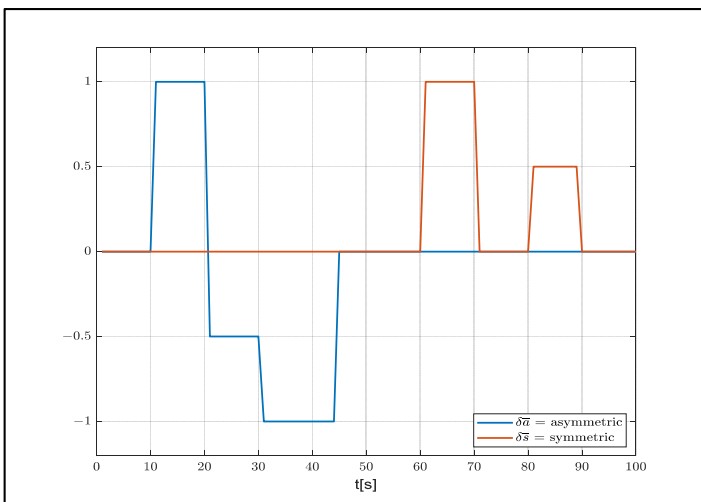

**Figure 2.** Recommended control input.

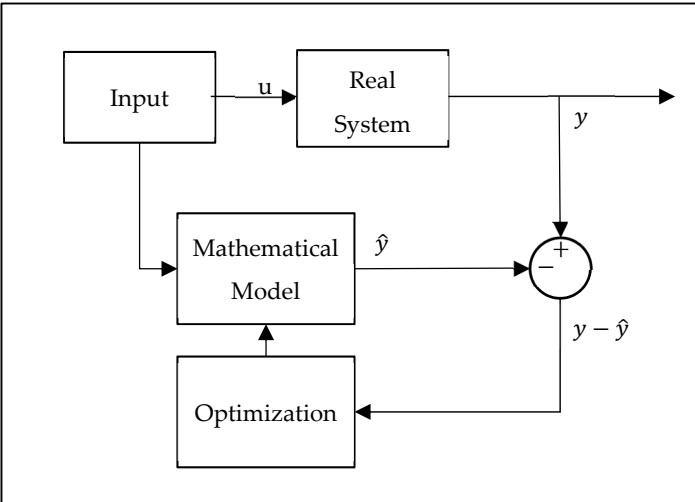

**Figure 3.** Output-error method.

The output-error method is the most widely preferred method for parameter estimation in the literature. In his study, Grauer calculated a dynamic model of an aircraft during flight by adapting the output-error method, which is usually carried out using post-flight data, to real-time flight data [4]. In another study using the output-error method, Jann

estimated the state variables of a parachute landing system called ALEX via sensor inputs (GPS, Magnetometers, Gyros, Accelerometers) [5]. On the other hand, Jaiswal, Prakash, and Chaturvedi estimated the aerodynamic coefficients of a parachute landing system using the maximum likelihood method and the output-error method [6].

In addition to statistical methods, machine learning techniques, which are increasing in popularity day by day, have also been successfully used in solving system identification problems. In the literature, artificial neural networks have been used in modeling aircraft dynamics [7–11], estimating aerodynamic forces and moments [12–15], and in controller designs [16,17]. Both feed-forward neural networks [14,18] and recurrent neural networks have been widely used in these studies [19]. Roudbari and Saghafi proposed a new method for describing the dynamics of highly maneuverable aircraft. In the model they developed, they modeled the flight dynamics with artificial neural networks. The difference between their approach and those of traditional methods is that they did not use aerodynamic information during the training process [20]. Bagherzadeh supported a model with flight dynamics in order to increase the performance of the artificial neural network model [21].

The development of deep learning methods has enabled these methods to be used frequently in system identification problems. The residual neural network approach, which is one type of deep neural network, is one of the methods used to solve these problems. Goyal and Benner developed a special architecture for dynamic systems called LQResNET [22]. The method they proposed allowed for the use of observations in the modeling of dynamical systems. Their model was based on the principle that the rate of a variable depends on the linear and quadratic forms of the variable. Chen and Xiu suggested the framework called gResNet. They defined the residual as the estimation error of the prior model. They also used a DNN to model the residual [23].

In this study, a NARX Network with a serial-parallel structure was used to identify an unknown aerial delivery system with a ram-air parachute. The dataset was created using the software-in-the-loop method (software in the loop). Gazebo was used as the simulator and PX4 was used as the autopilot software. The performance of the NARX network differed according to parameters used, such as the selected training algorithm, the input and output delays, the hidden layer, and the number of neurons. Within the scope of this study, each parameter was examined independently. Models were trained using MATLAB 2020a.

## 2. Mathematical Model

In this study, a 6-degree-of-freedom model developed for a parachute landing system was used [24]. The Equations of motion of the vehicle can be written as:

$$mI_{3x3} \begin{bmatrix} \dot{u} \\ \dot{v} \\ \dot{w} \end{bmatrix} = F - mS(\omega) \begin{bmatrix} u \\ v \\ w \end{bmatrix}, \tag{1}$$

$$I \begin{bmatrix} \dot{p} \\ \dot{q} \\ r \end{bmatrix} = M - S(\omega)I \begin{bmatrix} p \\ q \\ r \end{bmatrix}, \tag{2}$$

where $m$ is the mass, $I$ is the inertia matrix, $[u, v\ w]$ are linear velocities, $[p, q\ r]$ are the angular velocities in the body frame, $S(\omega)$ is a skew-symmetric matrix consisting of linear velocity vectors, $F$ is the force, and $M$ is moment.

Due to the xz-symmetry plane of the parachute landing system, the inertial matrices consist of 4 unique components.

$$S(\omega) = \begin{bmatrix} 0 & -r & q \\ r & 0 & -p \\ -q & p & 0 \end{bmatrix} \tag{3}$$

$$I = \begin{bmatrix} I_{xx} & 0 & I_{xz} \\ 0 & I_{yy} & 0 \\ I_{xz} & 0 & I_{zz} \end{bmatrix} \tag{4}$$

The forces and moments affecting the parachute are caused by gravity and aerodynamic forces. The gravitational force can be written according to the body (b) axis.

$$F_g = mg \begin{bmatrix} -\sin(\theta) \\ \cos(\theta)\sin(\Phi) \\ \cos(\theta)\cos(\Phi) \end{bmatrix} \tag{5}$$

The aerodynamic forces acting on the system are written using the relevant aerodynamic coefficients ($C_{D0}$, $C_{D\alpha^2}$, $C_{D\delta_s}$, $C_{Y\beta}$, $C_{L0}$, $C_{L\alpha}$, $C_{L\delta_s}$), according to the body axis.

$$F_a = QS_w^b R \begin{bmatrix} C_{D0} + C_{D\alpha^2}\alpha^2 + C_{D\delta_s}\bar{\delta}_s \\ C_{Y\beta}\beta \\ C_{L0} + C_{L\alpha}\alpha + C_{L\delta_s}\bar{\delta}_s \end{bmatrix} \tag{6}$$

In this Equation, $S$ represents the parachute surface area, $\bar{\delta}_s$ represents symmetric trailing edge deflection, and ($_w^b R$) is the rotation matrix from the aerodynamic coordinate system to the body axis.

$$_w^b R = R_\alpha R_\beta = \begin{bmatrix} \cos(\alpha) & 0 & -\sin(\alpha) \\ 0 & 1 & 0 \\ \sin(\alpha) & 0 & \cos(\alpha) \end{bmatrix} \begin{bmatrix} \cos(\beta) & \sin(\beta) & 0 \\ -\sin(\beta) & \cos(\beta) & 0 \\ 0 & 0 & 1 \end{bmatrix} \tag{7}$$

$$_w^b R = R_\alpha R_\beta = \begin{bmatrix} \cos(\alpha)\cos(\beta) & \cos(\alpha)\sin(\beta) & -\sin(\alpha) \\ -\sin(\beta) & \cos(\beta) & 0 \\ \sin(\alpha)\cos(\beta) & \sin(\alpha)\sin(\beta) & \cos(\alpha) \end{bmatrix} \tag{8}$$

The angle of attack and slip angle are obtained from the velocity vector in the body axis.

$$\alpha = \tan^{-1}\left(\frac{v_z}{v_x}\right) \tag{9}$$

$$\beta = \tan^{-1}\left(\frac{v_y}{\sqrt{v_x^2 + v_z^2}}\right) \tag{10}$$

The velocity vector in the body axis consists of the global velocity and the wind effect.

$$V_a = \begin{bmatrix} v_x \\ v_y \\ v_z \end{bmatrix} = \begin{bmatrix} u \\ v \\ w \end{bmatrix} - _n^b R \begin{bmatrix} w_x \\ w_y \\ w_z \end{bmatrix} \tag{11}$$

$_n^b R$ is the rotation matrix from the coordinate system on the North-East-Down-axis which has its origin in the center of mass of the parachute to the body axis. Euler angles (roll, pitch, yaw) are used in this notation.

$$R_\phi = \begin{bmatrix} 1 & 0 & 0 \\ 0 & \cos(\phi) & \sin(\phi) \\ 0 & -\sin(\phi) & \cos(\phi) \end{bmatrix} \tag{12}$$

$$R_\theta = \begin{bmatrix} \cos(\theta) & 0 & -\sin(\theta) \\ 0 & 1 & 0 \\ \sin(\theta) & 0 & \cos(\theta) \end{bmatrix} \tag{13}$$

$$R_\psi = \begin{bmatrix} 1 & 0 & 0 \\ 0 & \cos(\psi) & \sin(\psi) \\ 0 & -\sin(\psi) & \cos(\psi) \end{bmatrix} \tag{14}$$

$$_n^b R = R_\phi R_\theta R_\psi \tag{15}$$

Aerodynamic moments affecting the parachute can also be written using the relevant coefficients ($C_{l\beta}$, $C_{lp}$, $C_{lr}$, $C_{l\delta_a}$, $C_{m0}$, $C_{m\alpha}$, $C_{mq}$, $C_{n\beta}$, $C_{np}$, $C_{nr}$, $C_{n\delta_a}$). These are roll, pitch, and yaw moments, respectively [2].

$$M_a = \frac{\rho V_a^2 S}{2} \begin{bmatrix} b\left(C_{l\beta}\beta + \frac{b}{2V_a}C_{lp}p + \frac{b}{2V_a}C_{lr}r + C_{l\delta_a}\overline{\delta}_a\right) \\ \overline{c}\left(C_{m0} + C_{m\alpha}\alpha + \frac{c}{2V_a}C_{mq}q\right) \\ b\left(C_{n\beta}\beta + \frac{b}{2V_a}C_{np}p + \frac{b}{2V_a}C_{nr}r + C_{n\delta_a}\overline{\delta}_a\right) \end{bmatrix} \tag{16}$$

here, $\rho$ is air density, $\overline{c}$ represents mean aerodynamic chord, $\overline{\delta}_a = \delta_a / \delta_{a\,max}$ is asymmetric trailing-edge deflection, and $S$ is the canopy reference area.

## 3. Materials and Methods

The dataset was created using the software-in-the-loop method (software in the loop). Gazebo was used as the simulator and PX4 was used as the autopilot software. A virtual flight was performed in the Gazebo environment (Figure 4).

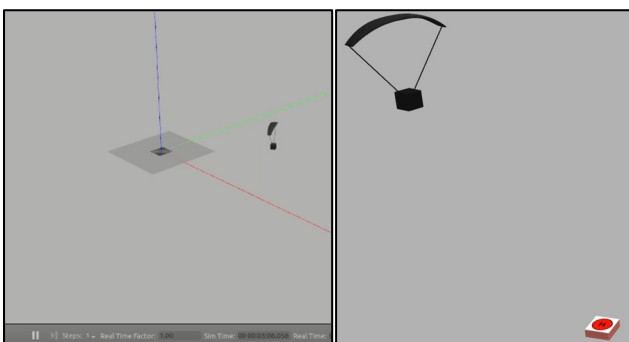

**Figure 4.** Gazebo simulation of the system.

The parameters required for the simulation were used considering the autonomous landing system with a parachute model named Snowflake (Table 1) [3].

**Table 1.** Parameters of the Snowflake parachute model [3].

| Parameter | Value |
|---|---|
| Mass ($m$) | 1.9 kg |
| Canopy reference area ($S$) | 1 m$^2$ |
| Inertia matrix ($I$) | $\begin{bmatrix} 0.042 & 0 & 0.0068 \\ 0 & 0.027 & 0 \\ 0.0068 & 0 & 0.054 \end{bmatrix}$ |
| Maximum brake deflection ($\delta_{smax}$) | 0.25 m |
| Aerodynamic coefficients | $\begin{matrix} C_{D0} = 0.15 & C_{D\alpha^2} = 0.90 \\ C_{Y\beta} = -0.05 & C_{L0} = 0.25 \\ C_{L\alpha} = 0.68 & C_{m0} = 0 \\ C_{m\alpha} = 0 & C_{mq} = -0.265 \\ C_{l\beta} = -0.036 & C_{lp} = -0.355 \\ C_{lr} = 0 & C_{l\delta_a} = 0.15 \\ C_{n\beta} = -0.036 & C_{np} = 0 \\ C_{D0} = -0.09 & C_{n\delta_a} = 0.003 \end{matrix}$ |

Gazebo compatible sensor models were used to obtain the flight data for the vehicle in the simulation environment. These consisted of a gyroscope, magnetometer, accelerometer, barometer, and GPS. The estimation of the state variables of the vehicle was carried out with PX4 software, using the extended Kalman filter.

PX4 has a state estimation module called EKF2 which uses the EKF algorithm. It uses IMU data in the state prediction phase. To correct these values, a GPS and barometer are used in the state correction phase [24].

Simplified models of the sensors used can be shown similarly [25]:

$$x_m = x + b + n, \tag{17}$$

$$\dot{b} = n_b, \tag{18}$$

where $x_m$ is the measured value; $x$ is the real value; and $b$, $n$, and $n_b$ represent bias and Gaussian noise, respectively. The sensor parameters can also be expressed using this notation. The sensor parameters used in the simulation are given in Table 2.

**Table 2.** Parameters used in the simulation.

| Sensors | Noise Density ($\sigma_n$) | Random Walk ($\sigma_{n_b}$) | Bias Correlation Time ($\sigma_n$) |
|---|---|---|---|
| Gyroscope | 0.00018 | 0.000 | 1000.0 |
| Accelerometer | 0.00186 | 0.006 | 300.0 |
| Magnetometer | 0.00040 | 0.000 | 600.0 |

The simulation was carried out in a windless environment and the air density was 1.225 kg/m$^3$. In the simulation, the system was released from a height of 500 m. Dropping occurred in 30 s. Control inputs $\overline{\delta}_a$ and $\overline{\delta}_s$ are given as full right and full left (Figure 5).

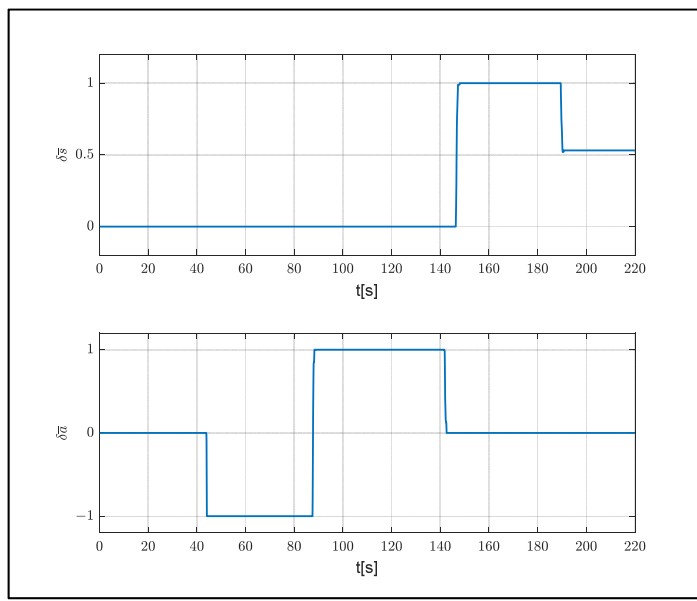

**Figure 5.** Control inputs.

The flight data received from the system were arranged and the input vector $x$ and the output vector y were created.

$$x = [\partial \overline{a} \; \partial \overline{s}] \tag{19}$$

$$y = [u \; v \; w \; p \; q \; r] \tag{20}$$

A total of 270 s of data were reduced to 225 s to cover the flight section, and 2250 pieces of data were produced using a 10 Hz measurement. The position and velocity of the vehicle in the flight data used are shown in Figures 6–8.

In order to improve the performance of the model, 70% of the flight was used for training and the remaining 30% was used in the testing process. Since the landing position is the most important phase, the first phase of the flight was selected as the training data.

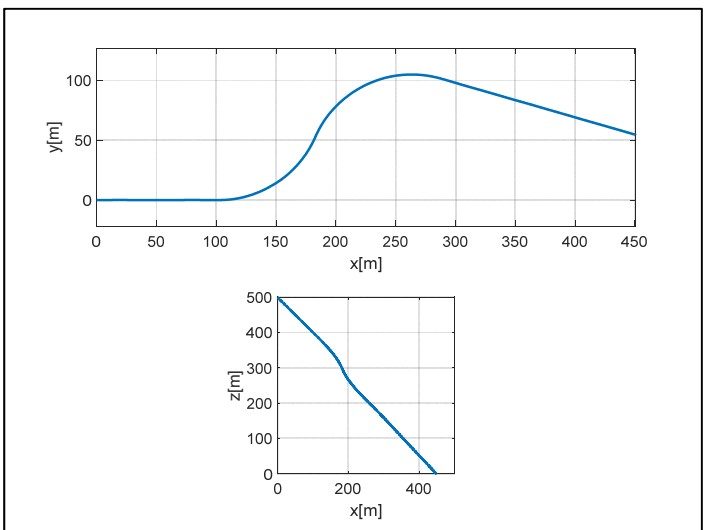

**Figure 6.** Position of the system.

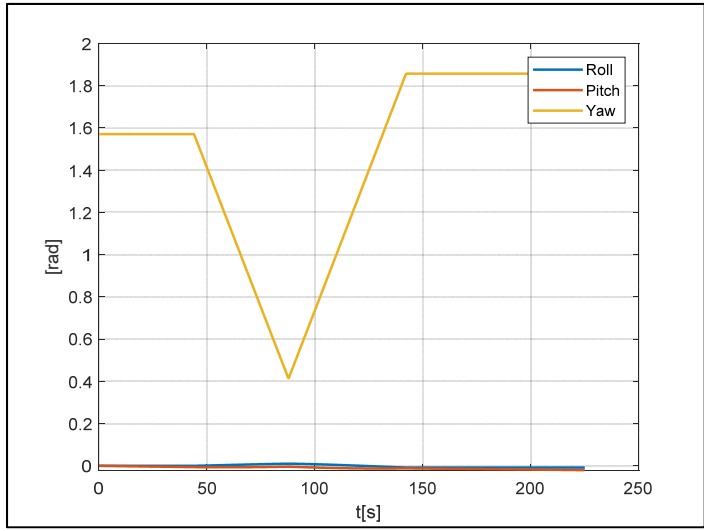

**Figure 7.** Euler angles.

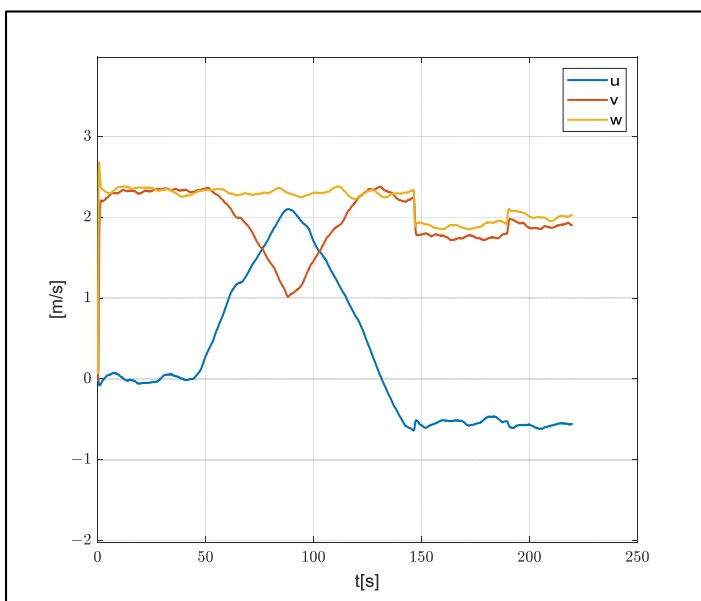

**Figure 8.** Velocity of the system.

## 3.1. NARX Network

A nonlinear autoregressive exogenous (NARX) network is a nonlinear model representation used in time series models. In this notation, the model's outputs depend on the past output values, the inputs, and the past values of the inputs. Its mathematical expression is given as follows:

$$y(t) = f\big[y(t-1),\, y(t-2),\, \ldots,\, y(t-n_y); u(t),\, u(t-1),\, \ldots,\, u(t-n_u)\big], \quad (21)$$

where $y$ denotes outputs, $u$ denotes inputs, and $f$ represents a nonlinear function. The structure in which f is modeled as a neural network is named the NARX neural network (NARX network) [26]. This model has been used for modeling conventional fixed-wing [27,28] and rotary-wing [29,30] aircraft. A NARX neural network can be modeled using two types of models: parallel and serial-parallel (Figure 9). In the parallel model, the estimated output values are fed back into the system.

$$\hat{y}(t) = f\big[\hat{y}(t-1),\, \hat{y}(t-2),\, \ldots,\, \hat{y}(t-n_y); u(t),\, u(t-1),\, \ldots,\, u(t-n_u)\big] \quad (22)$$

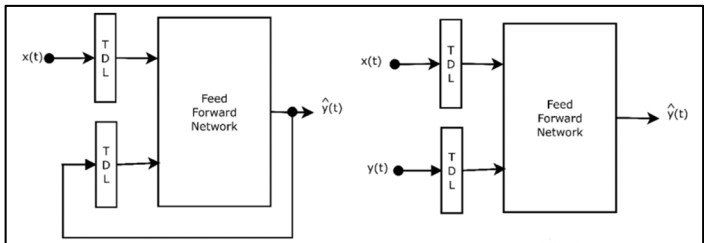

**Figure 9.** Parallel (**Left**) and serial-parallel (**Right**) NARX networks.

In the serial-parallel model, only real system outputs are used:

$$\hat{y}(t) = f\big[y(t-1),\, y(t-2),\, \ldots,\, y(t-n_y); u(t),\, u(t-1),\, \ldots,\, u(t-n_u)\big] \quad (23)$$

where $\hat{y}(t)$ represents the estimated output value time $t$.

Since the data set used in this study included real system outputs, the serial-parallel structure was preferred. The feed-forward network block shown in Figure 9 consisted of multilayer feedforward neural networks, which consisted of at least one hidden layer

and neurons. Each neuron calculated the outputs with the help of the activation function, determined using the inputs and their weights, as shown in Figure 10, where, $x_n$, $w_n$, $b$, and $f$ represent inputs, weights, bias, and the activation function, respectively. The architecture of the NARX neural network with a serial-parallel structure is shown in Figure 11.

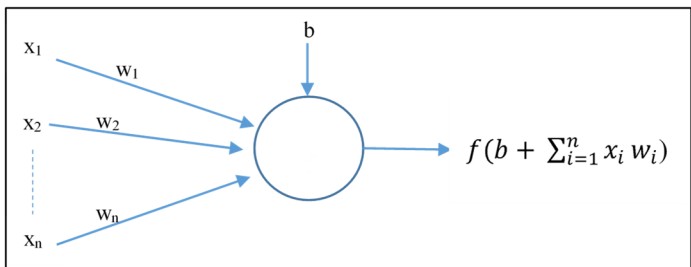

**Figure 10.** Structure of a neuron.

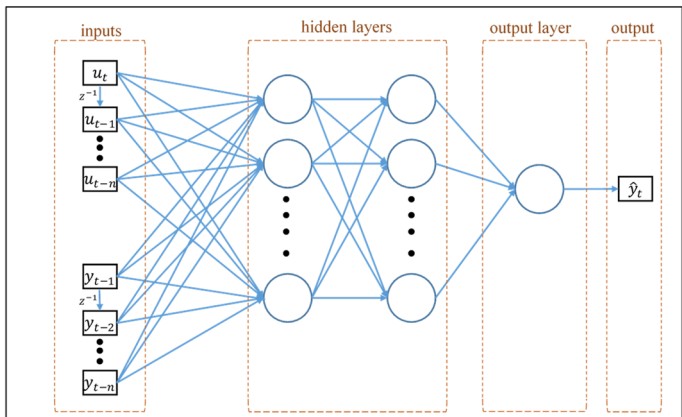

**Figure 11.** Serial-parallel NARX network architecture.

The selection of the activation functions plays an important role in the model design. The functions used in the hidden layers and the functions used in the output layer vary. Differentiable functions are preferred in hidden layers. These functions, which are preferred over linear functions during training, enable the models to perform successfully with more complex problems. In the literature, functions that are frequently used in hidden layers are ReLU (Rectified Linear Activation), sigmoid (logistic), and Tanh (hyperbolic tangent) functions. The function used in the output layer differs according to the type of problem. Linear functions are used in regression problems, whereas softmax or sigmoid functions are used in classification problems. This concept is illustrated in detail in Table 3.

The process of calculating and updating the weights is called training. The aim here is to minimize the targeted error function for model performance. In the neural network model, this function can be written as the sum of the squares of the errors:

$$E = \sum_{i=1}^{n} e_i^2, \tag{24}$$

where $e$ is the error and $n$ is the number of data.

**Table 3.** Activation functions.

| Function | | Plot |
|---|---|---|
| ReLU | $f(x) = \begin{cases} x, & x > 0 \\ 0, & x \leq 0 \end{cases}$ | 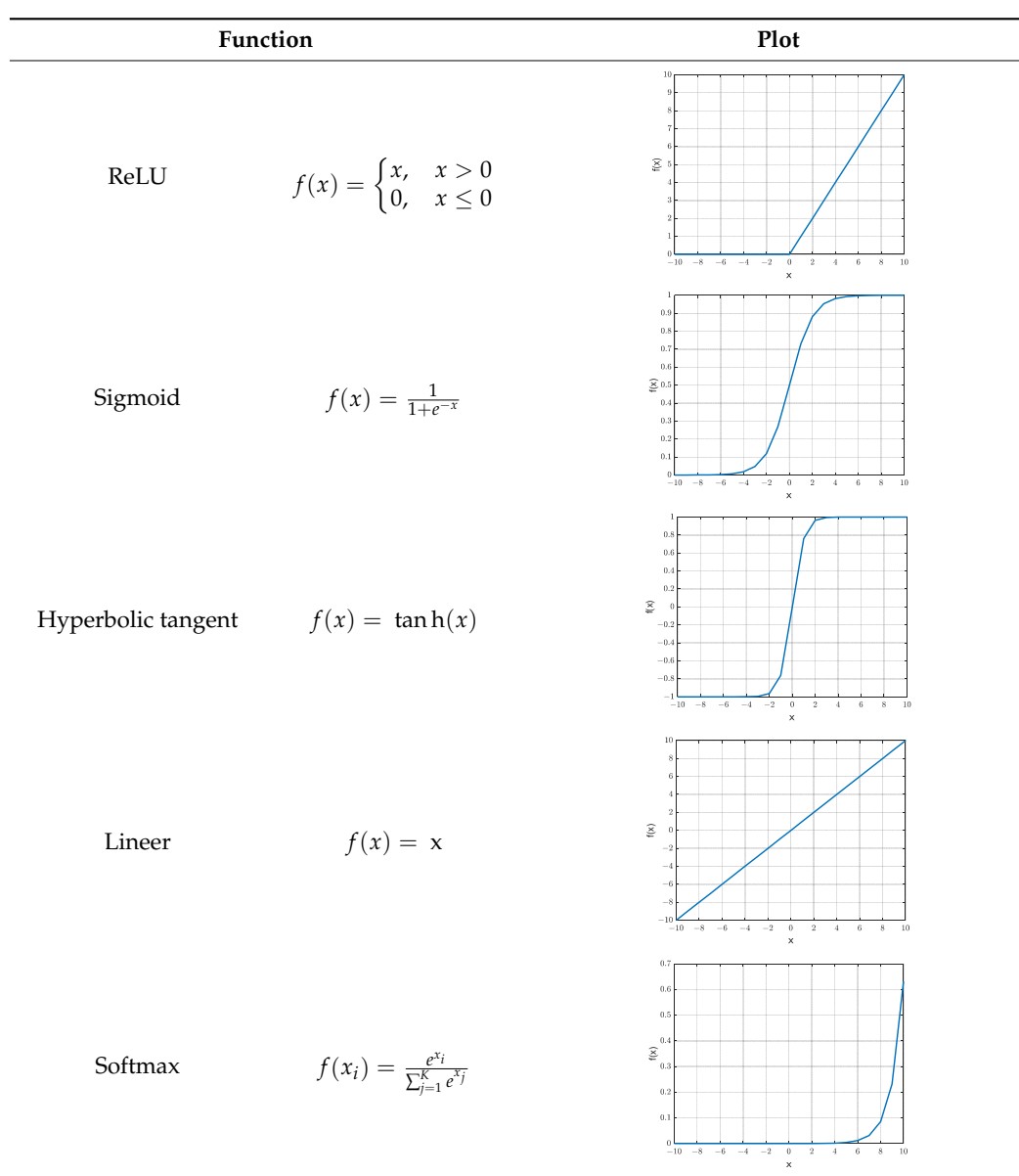 |
| Sigmoid | $f(x) = \frac{1}{1+e^{-x}}$ | |
| Hyperbolic tangent | $f(x) = \tan h(x)$ | |
| Lineer | $f(x) = x$ | |
| Softmax | $f(x_i) = \frac{e^{x_i}}{\sum_{j=1}^{K} e^{x_j}}$ | |

The training algorithm used in feed-forward neural network methods is known as the back-propagation algorithm [31]. Since the convergence rate of the steepest descent method, which is used as a standard in the back-propagation algorithm, is slow, many learning algorithms have been developed for neural network training. The main ones are the Levenberg–Marquardt algorithm [32], the Bayesian regularization algorithm [33], and the scaled conjugate gradient algorithm [34].

*3.2. Levenberg–Marquardt*

The Levenberg–Marquardt algorithm is a second-order training algorithm used in solving nonlinear optimization problems. According to the weight values that need to be updated, the Jacobian of the error function shown in Equation (23) can be calculated as follows:

$$J = \begin{bmatrix} \frac{\partial e_1}{\partial w_1} & \cdots & \frac{\partial e_1}{\partial w_m} \\ \vdots & \ddots & \vdots \\ \frac{\partial e_n}{\partial w_1} & \cdots & \frac{\partial e_n}{\partial w_m} \end{bmatrix}, \tag{25}$$

where $m$ is the number of weights in the network. After finding the Jacobian matrix, the gradient vector ($g$) and the Hessian matrix ($H$) can also be calculated.

$$g = J^T e \tag{26}$$

$$H = J^T J \tag{27}$$

The weights are updated based on the Jacobian matrix.

$$w_{i+1} = w_i - \left( J_i^T J_i + \alpha_i I \right)^{-1} \left( 2 J_i^T e_i \right) g = J^T e, \tag{28}$$

where $\alpha_i$ is the learning coefficient and $I$ is the unit matrix. A theoretical analysis can be found in [35].

*3.3. Bayesian Regularization*

The error function is rearranged using the regularization method to generalize the neural network [36]:

$$F = \mu E_w + \nu E, \tag{29}$$

where $\mu$ and $\nu$ are the regularization parameters and $E_w$ is the sum of the squared weights. The Bayesian regularization method is used for the optimization of the editing parameters. Considering the weight values as random variables, it aims to calculate the weight values that will maximize the posterior probability distribution of the weights in the given data set. The posterior distribution can be expressed according to the Bayes rule:

$$P(w|D, \mu, \nu, N) = \frac{P(D|w, \nu, N)\, P(w|\mu, N)}{P(D|\mu, \nu, N)}, \tag{30}$$

where $D$ represents the dataset and $N$ represents the neural network model. $P(D|w, \nu, N)$ expresses the likelihood function, $P(w|\mu, N)$ is the prior density, and $P(D|\mu, \nu, N)$ is the normalization factor. It can be said that the noise in the dataset and in the weights has a Gaussian distribution. Thus, the likelihood function and antecedent intensity values can be calculated.

$$P(D|w, \nu, N) = \frac{e^{-\nu E}}{Z(\nu)} \tag{31}$$

$$P(w|\mu, N) = \frac{e^{-\mu E_w}}{Z_w(\mu)} \tag{32}$$

Here, $Z = \left( \frac{\pi}{\nu} \right)^{n/2}$ and $Z_w = \left( \frac{\pi}{\mu} \right)^{m/2}$. By rearranging these equations, the posterior distribution to the weights can be rewritten.

$$P(w|D, \mu, \nu, N) = \frac{e^{-(\mu E_w + \nu E)}}{Z_w(\mu) Z(\nu)} \tag{33}$$

Regularization parameters are effective in the $N$ model. The Bayes rule can be applied for the optimization of these parameters.

$$P(\mu, \nu|D, N) = \frac{P(D|\mu, \nu, N)\, P(\mu, \nu|N)}{P(D|N)} \tag{34}$$

As can be seen in Equation (34), the function $P(D|\mu, \nu, N)$ is directly proportional to $P(\mu, \nu|D, N)$. Therefore, the maximum value of the function $P(D|\mu, \nu, N)$ must be calculated. Adjustment parameters can be calculated using the Taylor expansion of Equation (29). A theoretical analysis can be found in [37].

$$\mu = \frac{\gamma}{2E_w} \tag{35}$$

$$\nu = \frac{n - \gamma}{2E} \tag{36}$$

$$\gamma = m - \mu \, tr(H^{-1}) \tag{37}$$

*3.4. Scaled Conjugate Gradient*

In the steepest descent algorithm implemented in the standard back-propagation algorithm, a search is made in the opposite direction of the gradient vector while updating the weights. Although the error function decreases rapidly in this direction, the same cannot be said for the convergence rate. Conjugate gradient algorithms search using the direction with the fastest convergence. This direction is called the conjugate direction. In this method, the search first starts in the reverse of the gradient vector, similarly to the steepest descent algorithm. It differs from the second iteration as follows.

$$p_0 = -g_0 \tag{38}$$

$$x_{k+1} = x_k + \alpha_k g_k \tag{39}$$

$$p_k = -g_k + \beta_k p_{k-1} \tag{40}$$

Different algorithms have been developed according to the way in which the $\beta_k$ coefficient is calculated. Moller, on the other hand, combined the LM algorithm and the conjugate gradient algorithm for the calculation of the number of steps in the algorithm he developed. This algorithm is called the scaled conjugate gradient algorithm [35]. In this algorithm, which is based on calculating the approximate value of the Hessian matrix, the design parameters change in each iteration and are independent of the user. This is the most important factor affecting the success of the algorithm.

$$H_k = \frac{E'(w_k + \sigma_k p_k) - E'(w_k)}{\sigma_k} + \lambda_k p_k \tag{41}$$

$$\beta_k = \frac{\left(|g_{k+1}|^2 - g_{k+1}^T g_k\right)}{g_k^T g_k} \tag{42}$$

$$p_{k+1} = -g_{k+1} + \beta_k p_k \tag{43}$$

**4. Results and Discussion**

The performance of the NARX network differs according to parameters used, such as the selected training algorithm, the input and output delays, the hidden layer, and the number of neurons. Within the scope of this study, each parameter was examined independently. Models were trained using MATLAB 2020a. The root-mean-square error (RMSE) and mean absolute error (MAE) values were used to evaluate model performance. The metrics used are presented in Table 4.

**Table 4.** Metrics used in the evaluation of models.

| Measures | Equation | Description |
|---|---|---|
| Root-mean-square error | $RMSE = \sqrt{\frac{\sum_{i=1}^{n} e_i^2}{n}}$ | Low values indicate that the model was successful. |
| Mean absolute error | $MAE = \frac{\sum_{i=1}^{n} |e_i|}{n}$ | |

First, the performance of the training algorithms (Bayes arrangement, Levenberg–Marquardt, scaled conjugate gradient) in a model consisting of a single hidden layer and 15 neurons was compared. The input and output delay vectors were determined as in [12]. A hyperbolic tangent was used as the activation function in the hidden layer and a linear function was used in the output layer. The errors according to the training algorithms are shown in Table 5.

**Table 5.** Performance based on the training algorithms.

| Algorithm | RMSE | MAE | RMSE | MAE | RMSE | MAE |
|---|---|---|---|---|---|---|
| LM | 0.0007 | 0.0005 | 0.0026 | 0.0023 | 0.0016 | 0.0011 |
| BR | 0.0007 | 0.0005 | 0.0025 | 0.0021 | 0.0015 | 0.0011 |
| SCG | 0.0260 | 0.0081 | 0.0101 | 0.0018 | 0.0218 | 0.0037 |

Despite the fast training time, SCG performed worse than LM and BR. At this stage, the hidden layer and the number of neurons within it were changed and the results were examined and shown in Table 6. BR was used as the training algorithm.

**Table 6.** Performance based on the number of hidden layers and neurons.

| No | Hidden Layer | | | | Train | | Test | | Total | |
|---|---|---|---|---|---|---|---|---|---|---|
| | 1 | 2 | 3 | 4 | RMSE | MAE | RMSE | MAE | RMSE | MAE |
| 1 | 10 | - | - | - | 0.0007 | 0.0005 | 0.0025 | 0.0021 | 0.0015 | 0.0011 |
| 2 | 3 | - | - | - | 0.1538 | 0.0051 | 0.0018 | 0.0016 | 0.1256 | 0.0039 |
| 3 | 5 | 2 | - | - | 0.0208 | 0.0072 | 0.0318 | 0.0312 | 0.0250 | 0.0152 |
| 4 | 5 | - | - | - | 0.0008 | 0.0006 | 0.0012 | 0.0010 | 0.0010 | 0.0007 |
| 5 | 10 | 5 | - | - | 0.0007 | 0.0005 | 0.0056 | 0.0044 | 0.0033 | 0.0018 |
| 6 | 25 | - | - | - | 0.0006 | 0.0005 | 0.0018 | 0.0015 | 0.0012 | 0.0008 |
| 7 | 50 | - | - | - | 0.0006 | 0.0005 | 0.0023 | 0.0019 | 0.0014 | 0.0010 |
| 8 | 15 | 12 | - | - | 0.0007 | 0.0005 | 0.0027 | 0.0020 | 0.0017 | 0.0010 |
| 9 | 15 | 12 | 12 | - | 0.0009 | 0.0005 | 0.0015 | 0.0013 | 0.0012 | 0.0008 |
| 10 | 15 | 12 | 12 | 6 | 0.0009 | 0.0005 | 0.0016 | 0.0014 | 0.0012 | 0.0008 |

According to the angle of attack and the slip angle, it can be seen that model 4, which consisted of a single hidden layer and five neurons, showed the best performance. A comparison of the model results with the real system is shown in Figure 12.

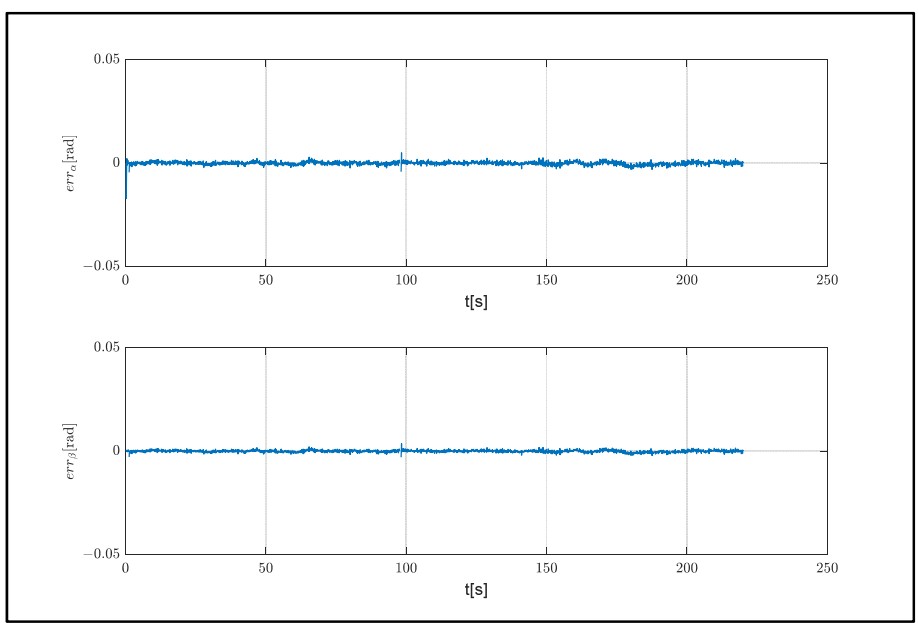

**Figure 12.** Estimation errors.

In order to observe the performance of the models with the same aerodynamic characteristics and different weights, the system weight was increased to 10 kg and a flight was carried out from an altitude of 1000 m. Control inputs produced during the flight are shown in Figure 13.

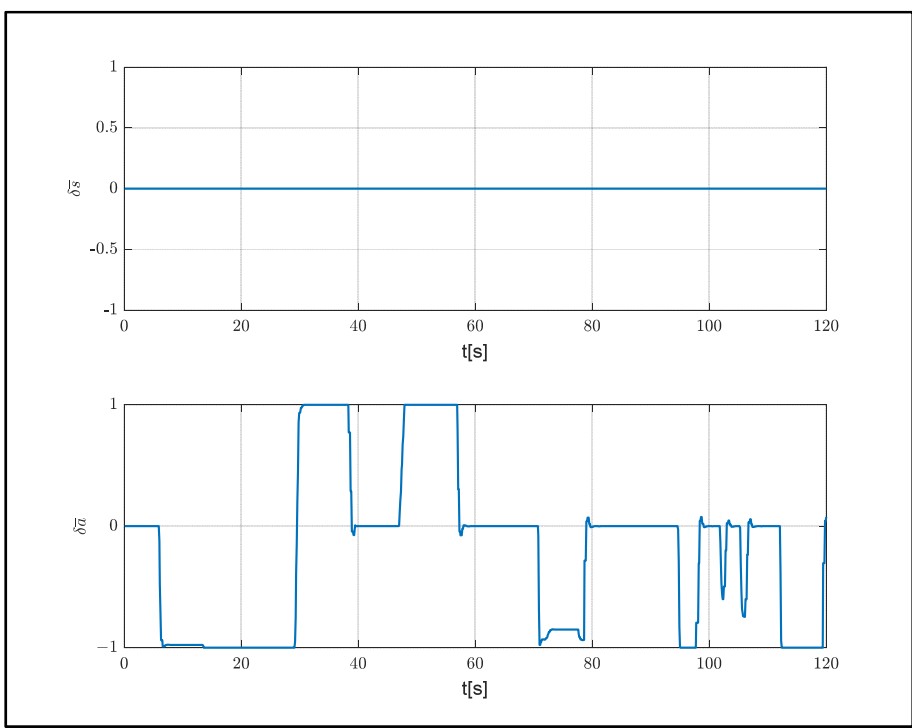

**Figure 13.** Control inputs for the 10 kg system.

The model performances for a 120 s flight were compared using error metrics and computational costs measures. As can be seen in Table 7, increasing the number of hidden layers and neurons increased the computation time. Considering the model performances, model number 4 exhibited the best performance.

**Table 7.** Model performances for increased weight.

| Model No | RMSE | MAE | Computational Cost (ms) |
|:---:|:---:|:---:|:---:|
| 1 | 0.2379 | 0.1782 | 42.274 |
| 2 | 0.2235 | 0.1523 | 40.971 |
| 3 | 0.2965 | 0.2206 | 49.747 |
| 4 | 0.1363 | 0.0973 | 43.325 |
| 5 | 0.3029 | 0.2376 | 43.777 |
| 6 | 0.2963 | 0.2071 | 43.872 |
| 7 | 0.2616 | 0.1802 | 44.537 |
| 8 | 0.3100 | 0.2182 | 66.263 |
| 9 | 0.3137 | 0.2219 | 66.851 |
| 10 | 0.3004 | 0.2244 | 66.961 |

The estimation errors for increased weight are shown in Figure 14. The increase in the number of hidden layers increased the overshoot values, although it did not result in any significant changes in model performance. Finally, the performance of the developed models was examined in a system with different aerodynamic properties. An aerial delivery system called ALEX was used to determine the necessary parameters (Table 8) [3].

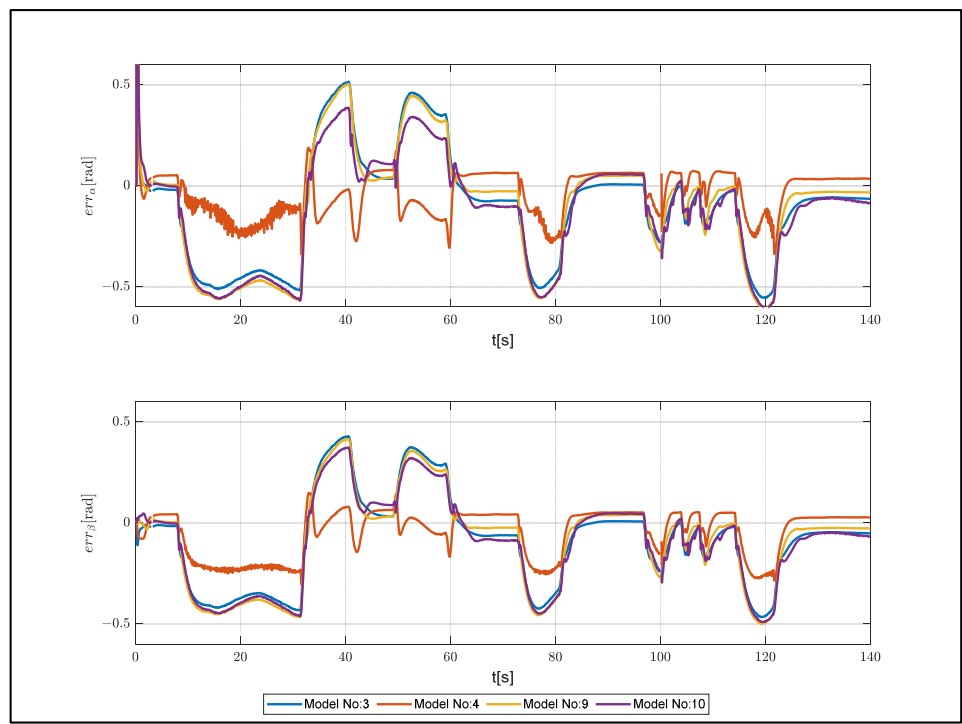

**Figure 14.** Estimation errors for increased weight.

**Table 8.** Parameters of ALEX [3].

| Parameter | Value |
|---|---|
| Mass ($m$) | 97.6 kg |
| Canopy reference area ($S$) | 19.72 m2 |
| Aerodynamic coefficients | $\begin{bmatrix} C_{D0} = 0.084 & C_{D\alpha^2} = 0.90 \\ C_{Y\beta} = -0.216 & C_{L0} = 0.25 \\ C_{L\alpha} = 2.36 & C_{m0} = 0 \\ C_{m\alpha} = 0 & C_{mq} = -0.174 \\ C_{l\beta} = 0.104 & C_{lp} = -0.149 \\ C_{lr} = 0.096 & C_{l\delta_a} = -0.048 \\ C_{n\beta} = 0.019 & C_{np} = -0.027 \\ C_{D0} = 0.084 & C_{n\delta_a} = 0.039 \end{bmatrix}$ |

The control inputs used in this flight, starting from a 200 m altitude, are shown in Figure 15.

**Table 9.** Model performances for ALEX.

| Model No | RMSE | MAE |
|---|---|---|
| 1 | 0.0296 | 0.0225 |
| 2 | 0.0491 | 0.0457 |
| 3 | 0.0781 | 0.0721 |
| 4 | 0.1251 | 0.1195 |
| 5 | 0.0413 | 0.0308 |
| 6 | 0.0562 | 0.0471 |
| 7 | 0.0388 | 0.0273 |
| 8 | 0.0734 | 0.0375 |
| 9 | 0.0510 | 0.0462 |
| 10 | 0.1024 | 0.0435 |

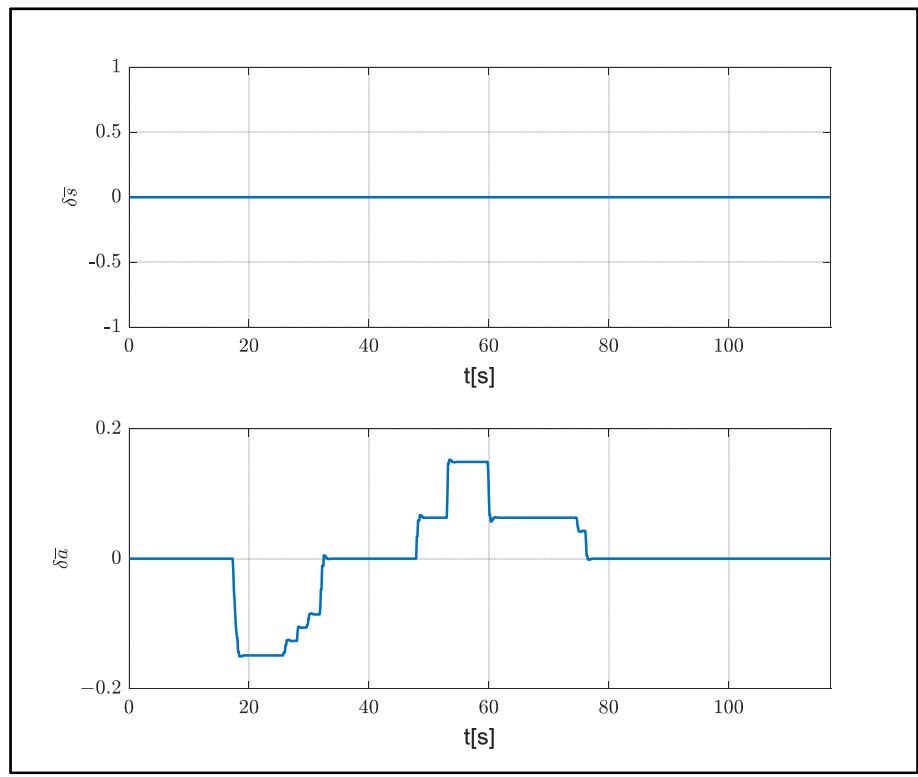

**Figure 15.** Control inputs for ALEX.

The performances of the models are shown in Table 9 and Figure 16. Model 1, which was found to have the best performance, consisted of a single hidden layer and 10 neurons.

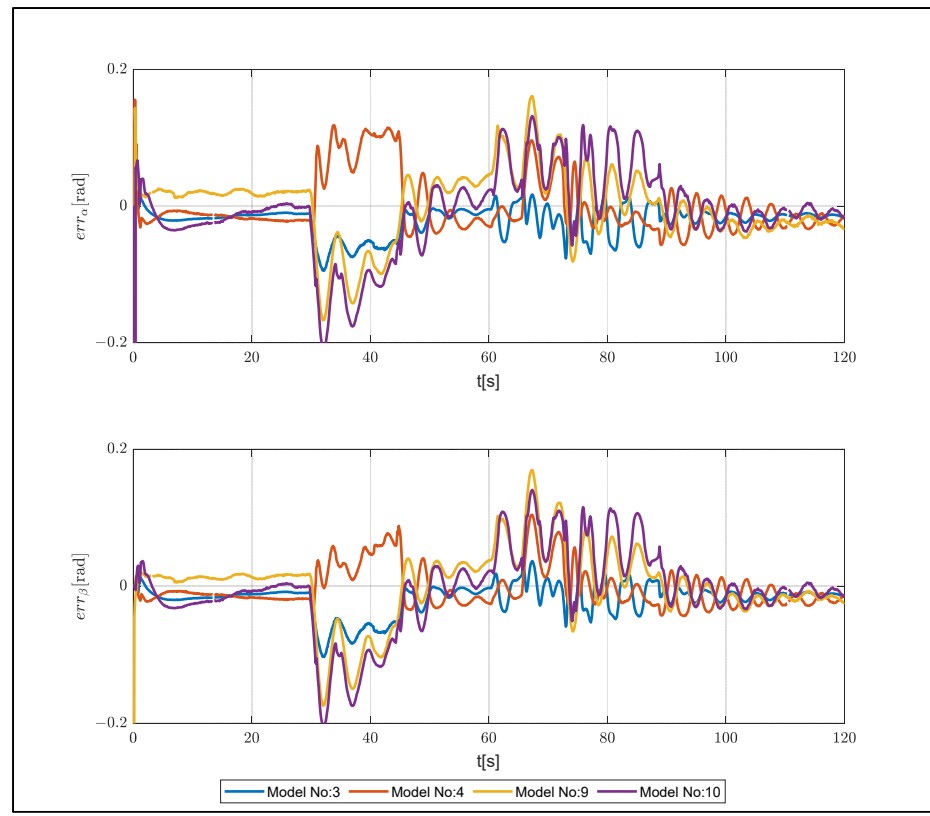

**Figure 16.** Estimation errors for ALEX.

In order to determine the limits of the developed models, the effects of weight and aerodynamic coefficients on the models were observed. The aerodynamic coefficients that determined the effects of control inputs on force and moment were chosen. Error rates were observed by changing the selected parameters by ±10%. RMSE was used as the error metric. As can be seen in Table 10, model 4, which consisted of a single hidden layer and five neurons, demonstrated the best performance.

**Table 10.** Model performances according to changes of 10%.

| No | m | | $C_{d\delta_s}$ | | $C_{l\delta_a}$ | | $C_{l\delta_s}$ | | $C_{n\delta_a}$ | |
|----|------|------|------|------|------|------|------|------|------|------|
| | +10% | −10% | +10% | +10% | −10% | +10% | −10% | −10% | −10% | −10% |
| 1 | 0.0035 | 0.0042 | 0.0036 | 0.0034 | 0.0033 | 0.0032 | 0.0030 | 0.0029 | 0.1345 | 0.0039 |
| 2 | 0.0070 | 0.0072 | 0.0063 | 0.0045 | 0.0057 | 0.0055 | 0.0052 | 0.0056 | 0.0397 | 0.0065 |
| 3 | 0.0156 | 0.0157 | 0.0129 | 0.0284 | 0.0166 | 0.0182 | 0.0179 | 0.0183 | 0.0591 | 0.0182 |
| 4 | 0.0032 | 0.0041 | 0.0032 | 0.0023 | 0.0031 | 0.0031 | 0.0027 | 0.0031 | 0.0739 | 0.0028 |
| 5 | 0.0055 | 0.0094 | 0.0072 | 0.0054 | 0.0064 | 0.0061 | 0.0053 | 0.0058 | 0.2064 | 0.0063 |
| 6 | 0.0123 | 0.0116 | 0.0111 | 0.0073 | 0.0099 | 0.0104 | 0.0081 | 0.0086 | 0.1334 | 0.0093 |
| 7 | 0.0119 | 0.0114 | 0.0122 | 0.0064 | 0.0086 | 0.0087 | 0.0073 | 0.0078 | 0.0623 | 0.0089 |
| 8 | 0.0123 | 0.0161 | 0.0027 | 0.0084 | 0.0121 | 0.0112 | 0.0098 | 0.0020 | 0.2058 | 0.0110 |
| 9 | 0.0120 | 0.0104 | 0.0099 | 0.0057 | 0.0110 | 0.0094 | 0.0076 | 0.0113 | 0.0578 | 0.0089 |
| 10 | 0.0121 | 0.0131 | 0.0102 | 0.0077 | 0.0112 | 0.0102 | 0.0090 | 0.0100 | 0.0950 | 0.0091 |

Considering the maximum error of five degrees, the limits of the models could be determined approximately, according to the parameters, via interpolation. The results are shown in Table 11.

**Table 11.** Limits of models.

| No | m | | $C_{d\delta_s}$ | | $C_{l\delta_a}$ | | $C_{l\delta_s}$ | | $C_{n\delta_a}$ | |
|----|------|------|------|------|------|------|------|------|------|------|
| | Max | Min | Max | Min | Max | Min | Max | Min | Max | Min |
| 1 | 6.623 | 0.000 | 0.342 | −0.156 | 0.545 | −0.258 | 0.390 | −0.200 | 0.0032 | −0.0037 |
| 2 | 4.261 | 0.000 | 0.238 | −0.093 | 0.379 | −0.087 | 0.267 | −0.055 | 0.0037 | −0.0010 |
| 3 | 2.960 | 0.847 | 0.167 | 0.069 | 0.229 | 0.078 | 0.149 | 0.052 | 0.0034 | 0.0016 |
| 4 | 7.066 | 0.000 | 0.372 | −0.278 | 0.571 | −0.271 | 0.422 | −0.181 | 0.0034 | −0.0063 |
| 5 | 4.905 | 0.141 | 0.221 | −0.061 | 0.354 | −0.064 | 0.264 | −0.050 | 0.0031 | −0.0011 |
| 6 | 3.244 | 0.475 | 0.178 | −0.019 | 0.282 | 0.025 | 0.207 | −0.001 | 0.0032 | 0.0002 |
| 7 | 3.289 | 0.450 | 0.171 | −0.036 | 0.302 | 0.000 | 0.219 | −0.012 | 0.0034 | 0.0001 |
| 8 | 3.244 | 0.873 | 0.422 | −0.004 | 0.258 | 0.033 | 0.189 | −0.335 | 0.0031 | 0.0006 |
| 9 | 3.278 | 0.311 | 0.188 | −0.053 | 0.269 | 0.011 | 0.214 | 0.023 | 0.0035 | 0.0001 |
| 10 | 3.266 | 0.638 | 0.185 | −0.013 | 0.267 | 0.022 | 0.197 | 0.013 | 0.0033 | 0.0001 |

## 5. Conclusions

In this study, a simulation environment was designed for a parachute landing system in the Gazebo/ROS environment. By implementing an aerial delivery system in PX4 autopilot software, the necessary infrastructure for a software-in-the-loop system was created. Flights were performed in the simulation environment and flight data were collected. Using these data for the description of the system, an NARX network model was trained, and a dynamic model was used to estimate the system. During the training process, different training algorithms were used (LM, BR, and SCG) and the effects of the numbers of hidden layers and neurons were observed. The effects of weight and aerodynamic coefficients on the models were also examined and the model limits were determined. As a result of the examinations, the model consisting of a single hidden layer and five neurons outperformed the other models evaluated. As the rates of different model parameters increase, the model which has the best performance may change. Therefore, errors in models can be improved by means of online training methods.

In future studies, pre-trained models will be updated using online training methods. Furthermore, the trained model will be tested using real flight data. After the model is verified, controller studies will be carried out and autonomous landing of the landing system will be carried out at the desired target location.

**Author Contributions:** Conceptualization, K.G. and A.T.Ş.; methodology, K.G.; software, K.G.; validation, K.G. and A.T.Ş.; investigation, K.G.; writing—original draft preparation, K.G.; writing—review and editing, A.T.Ş. All authors have read and agreed to the published version of the manuscript.

**Funding:** This research received no external funding.

**Institutional Review Board Statement:** Not applicable.

**Informed Consent Statement:** Not applicable.

**Data Availability Statement:** Not applicable.

**Conflicts of Interest:** The authors declare no conflict of interest.

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
