# Peer review of "System Identification of an Aerial Delivery System with a Ram-Air Parachute Using a NARX Network"

_aerospace, doi:10.3390/aerospace9080443_

Round 1

Reviewer 1 Report

The paper is a well written piece of work, however some issues shall be pointed:

  1. The model does not consider any wind of turbulence. This may seem unrelevant but it is critical from the point of view of a realistic application of the method. With no wind, the inertial sensors perfectly captures the dynamics of the system, so the parameter stimation process is straightforward.
  2. The values for the estimated parameters shall be indicated and discussed somewhere.

Author Response

Dear reviewer,

Thank you for the positive feedback and helpful comments. We provide a document consists our responses.

Sincerely

Reviewer 2 Report

In this study, NARX Network with serial-paralllel structure is used to identify the unknown Aerial Delivery System with the Ram-Air Parachute. However, in my opinion, several concerns listed as follows should be clarified or revised.

  1. I think the model size/Flops/speed are also very important forSystem Identification of an Aerial Delivery System.
  2. The authors should highlight the limitations of the proposed method.
  3. The analysis regarding performance improvement of the proposed method is not so convincing.
  4. The design details of each proposed module are elaborated in section 3. Unfortunately, there is no corresponding theoretical analysis to support the rationality of the design.
  5. Related work is a bit inadequate. For example,HFNet, RLLNet, etc.

Author Response

(The authors gave the same response as above.)

Round 2

Reviewer 1 Report

Noise in the instruments cannot reflect the fact that wind decouple measurements and vehicle performance.

Author Response

(The authors gave the same response as above.)

Reviewer 2 Report

I recommend accepting this paper.

Author Response

(The authors gave the same response as above.)
